# Blastoid B-Cell Neoplasms: Diagnostic Challenges and Solutions

**DOI:** 10.3390/cancers15030848

**Published:** 2023-01-30

**Authors:** Lianqun Qiu, Sa A. Wang, Guilin Tang, Wei Wang, Pei Lin, Jie Xu, C. Cameron Yin, Mahsa Khanlari, L. Jeffrey Medeiros, Shaoying Li

**Affiliations:** 1Department of Hematopathology, The University of Texas MD Anderson Cancer Center, Houston, TX 77030, USA; 2Department of Laboratory Medicine & Pathology, University of Washington, Seattle, WA 98115, USA; 3St. Jude Children’s Research Hospital, Memphis, TN 38105, USA

**Keywords:** high grade B-cell lymphoma, blastoid morphology, B-lymphoblastic leukemia/lymphoma, six-point scoring system, three-point immunohistochemistry scoring system, transcriptome profile, gene expression signature

## Abstract

**Simple Summary:**

Distinguishing blastoid HGBL from B-ALL can be challenging. We previously developed six-point flow cytometry-focused and three-point immunohistochemistry-focused scoring systems to aid in differential diagnosis. However, the six-point scoring system was derived from bone marrow cases and occasional cases may have a misleading score using either system. Herein, we assessed 121 blastoid-HGBL cases in comparison with 47 B-ALL cases enriched for CD34-negative neoplasm to validate the six-point scoring system in all tissue types and to compare the two scoring systems. The six-point scoring system showed a sensitivity of 100% in distinguishing HGBL versus B-ALL of any tissue type. The two scoring systems had a concordance score rate of 88% in blastoid HGBL and B-ALL. Thirteen cases showed misleading scores, including five HGBL and eight B-ALL, and the diagnosis was further validated by gene transcriptome profiling. Simultaneous employment of both scoring systems improved the accuracy of classification of blastoid B-cell neoplasms to 99%. Therefore, we suggest to use both scoring systems together to improve the accuracy of classification of blastoid B-cell neoplasms. Cases with discordant scores between the two scoring systems are extremely challenging neoplasms to classify that would require correlation with all available clinical and genetic features.

**Abstract:**

Blastoid B-cell neoplasms mainly include B-lymphoblastic leukemia/lymphoma (B-ALL), blastoid mantle cell lymphoma, and high-grade B-cell lymphoma with blastoid morphologic features (blastoid HGBL). Distinguishing blastoid HGBL from B-ALL can be challenging and we previously developed six-point flow cytometry-focused and three-point immunohistochemistry-focused scoring systems to aid in differential diagnosis. However, the six-point scoring system was derived from bone marrow cases and occasional cases may have a misleading score using either system. In this study, we assessed 121 cases of blastoid-HGBL (37 BM and 84 extramedullary) to validate the six-point scoring system in all tissue types and to further compare the two scoring systems. Compared with 47 B-ALL cases enriched for CD34-negative neoplasm, the 121 blastoid-HGBL cases showed distinctive pathologic features. The six-point scoring system showed a sensitivity of 100%. A comparison of the two scoring systems in blastoid HGBL (*n* = 64) and B-ALL (*n* = 37) showed a concordance score rate of 88%. Thirteen cases showed misleading scores, including five HGBL and eight B-ALL, and the diagnosis was further validated by gene transcriptome profiling. Twelve of thirteen cases had discordant scores between the two scoring systems. Simultaneous employment of both scoring systems improved the accuracy of classification of blastoid B-cell neoplasms to 99%. In conclusion, the previously defined six-point scoring system showed an excellent performance regardless of the tissue origin. Using both scoring systems together improves the accuracy of classification of blastoid B-cell neoplasms. Cases with discordant scores between the two scoring systems were extremely challenging neoplasms and classification required correlation with all available clinical and genetic features.

## 1. Introduction

Blastoid B-cell neoplasms mainly include B-lymphoblastic leukemia/lymphoma (B-ALL), blastoid variant mantle cell lymphoma (MCL), and high-grade B-cell lymphoma with blastoid morphologic features (blastoid HGBL). Blastoid MCL is relatively easier to recognize based on its immunophenotypic and cytogenetic features. The differential diagnosis between blastoid HGBL and CD34 negative B-ALL can be very challenging, especially when a case of blastoid HGBL presents with some unusual features, such as a leukemic presentation with dim CD45 and/or lack of surface light chain expression [1,2,3,4] or when a case of B-ALL lacks CD34 and TdT expression.

We previously evaluated 31 cases of blastoid HGBL presented in bone marrow and compared their clinicopathologic features with 36 cases of B-ALL [4]. In that study, we developed a six-point scoring system based on flow cytometric analysis of five markers (CD45, CD10, CD20, CD38, and TdT) together with MYC expression by immunohistochemistry and/or *MYC*-rearrangement by fluorescence in situ hybridization (FISH) and a diagnostic algorithm to aid in the differential diagnosis of blastoid B-cell neoplasms. This scoring system had a 100% sensitivity and 94% specificity. However, this scoring system was developed in bone marrow specimens and largely dependent on complicated flow cytometry examination.

As flow cytometry immunophenotypic analysis may not be always available, recently we compared 64 cases of blastoid HGBL versus B-ALL and developed a three-point immunohistochemistry (IHC)-based scoring system and a diagnostic algorithm to aid in the differential diagnosis of blastoid B-cell neoplasms [5]. This three-point system included three markers detected by immunohistochemistry, BCL6, TdT, and MYC; *MYC* also could be detected by FISH for rearrangement. A score of two or three supports a diagnosis of blastoid HGBL, whereas a score of one supports B-ALL. This scoring system had a sensitivity and specificity both of 92%.

In this study, we collected 121 cases of blastoid HGBL involving any tissue sites and assessed their clinicopathologic features in comparison with 47 cases of B-ALL. The aims of the study are to: (1) validate the previously reported six-point flow cytometry-focused scoring system in a larger cohort of cases involving bone marrow or extramedullary anatomic sites; (2) compare the two scoring systems and sort out extremely challenging cases and potential solutions for establishing the diagnosis; (3) validate the cases with discordant scores between the two scoring systems at gene expression level.

## 2. Materials and Methods

### 2.1. Case Selection

The archives of the Department of Hematopathology at The University of Texas MD Anderson Cancer Center were searched for high-grade B-cell lymphomas with blastoid morphology diagnosed between 1 January 2012 and 31 December 2021. We applied the inclusion and exclusion criteria described previously with modifications [4]. Briefly, inclusion criteria included HGBL cases with blastoid morphology involving any tissue site. Blastoid morphology in this study was defined as medium-size lymphoma cells with a scant amount of cytoplasm, finely dispersed nuclear chromatin, and inconspicuous nucleoli or small distinct nucleoli, if present, resembling lymphoblasts in hematoxylin and eosin-stained tissue sections, touch preparations, or aspirate smears if available [6]. Blastoid morphology was reviewed and confirmed by at least two pathologists independently in a blinded manner. The exclusion criteria were: (1) patients with a diagnosis of Burkitt lymphoma, B-ALL, blastoid variant of mantle cell lymphoma (including cyclin D1-negative cases), and blastoid follicular lymphoma; (2) cases with processing artifact that precluded adequate morphologic assessment; (3) cases in which *MYC* rearrangement status was not available. Forty-seven cases of B-ALL, diagnosed during the same time interval, were included as a comparison group. The diagnosis of blastoid HGBL and B-ALL was made according to the criteria defined in the 4th WHO Classification [6] and, for cases difficult to classify based on WHO criteria, the diagnosis was further confirmed by the gene transcriptome profile. For the purpose of this study, a disproportionally large subset of CD34-negative B-ALL (*n* = 25) cases were included. For the purpose of the study, previously reported cases were included [4,5]. The corresponding medical records were reviewed to obtain clinical information.

### 2.2. Immunophenotypic Analysis

Flow cytometric immunophenotyping was performed on cell suspensions of tissue biopsy specimens or bone marrow aspirates as described previously [7]. Erythrocytes were lysed with ammonium chloride (Pharm Lyse TM, BD Biosciences, San Diego, CA, USA) at room temperature for 10 min using a standard lyse/wash technique after incubation with monoclonal antibodies for 10 min at 4 °C. The following antibodies were used: CD2, cytoplasmic CD3, surface CD3, CD4, CD5, CD7, CD10, CD11c, CD13, CD14, CD15, CD19, CD20, CD22, CD23, CD25, CD33, CD34, CD36, CD38, CD41, CD44, CD45, CD49d, CD52, CD56, CD64, CD66c, cytoplasmic CD79a, CD81, CD117, CD123, CD200, cytoplasmic IgM, human leukocyte antigen (HLA)-DR, kappa, lambda, myeloperoxidase, and terminal deoxynucleotidyl transferase (TdT). All antibodies were purchased from BD Biosciences. Samples were acquired on FACSCanto II or FACSCalibur cytometers (Becton-Dickinson Biosciences, San Jose, CA, USA) and the data were analyzed using FCS Express software (De Novo Software, Los Angeles, CA, USA).

Immunohistochemical studies were performed using formalin-fixed paraffin-embedded (FFPE) tissue sections at the time of diagnosis or retrospectively for this study as described previously [7]. Antibodies specific for the following antigens were used: CD3, CD20, BCL6, and Ki-67 (Dako North America, Carpinteria, CA, USA); cyclin D1 (Thermo Scientific, Fremont, CA, USA); CD34 and PAX5 (BD Biosciences, San Jose, CA, USA.); CD10, BCL2, MUM1/IRF4, and TdT (Leica Biosystems, Buffalo Grove, IL, USA) and MYC (Ventana, Tucson, AZ, USA). Positive cutoff values used for evaluation were ≥30% for CD10, BCL6, ≥40% for MYC, and ≥50% for BCL2 [4].

### 2.3. Conventional Cytogenetic Analysis

Conventional cytogenetic analysis was performed on G-banded metaphase cells prepared from bone marrow aspirates, peripheral blood, or cell suspensions from tissue biopsy specimens, as described previously [8]. Twenty metaphases were analyzed and the results were reported using the 2020 International System for Human Cytogenetic Nomenclature [9]. A complex karyotype was defined as ≥3 numerical and structural abnormalities.

### 2.4. Fluorescence In Situ Hybridization

Fluorescence in situ hybridization (FISH) analysis was performed on bone marrow or cultured cells or formalin-fixed paraffin-embedded (FFPE) tissue sections according to the manufacturer’s instructions. The following FISH probes were used in the study: LSI *IGH/CCND1* dual-color dual-fusion probe; LSI *MYC* and *BCL6* dual-color break-apart probes; LSI *IGH/BCL2* dual-color dual-fusion probe (Vysis/Abbott Laboratories, Des Plaines, IL, USA). A total of 200 interphase nuclei were analyzed [10]. The cutoffs for considering a tumor sample positive for *MYC*, *BCL2*, and *BCL6* alterations were different in bone marrow smears versus FFPE tissue sections. However, these cutoffs were low (all ≤5%) and all cases in this study with *MYC*, *BCL2,* and/or *BCL6* abnormalities had abnormal signals present in >20% of all nuclei assessed.

### 2.5. Targeted Next Generation Sequencing

Gene mutation analysis was performed using DNA extracted from bone marrow aspirate specimens in a small subset of patients. Amplicon-based next generation sequencing (NGS) targeting the entire coding regions of a panel of 81 genes was performed using a MiSeq platform (Illumina, San Diego, CA, USA) to detect somatic mutations and insertions and/or deletions as previously described [11]. The 81-gene panel included: *ANKRD26*, *ASXL1*, *ASXL2*, *BCOR*, *BCORL1*, *BRAF*, *BRINP3*, *CALR*, *CBLB*, *CBLC*, *CBL*, *CRLF2*, *CREBBP*, *CEBPA*, *CSF3R*, *CUX1*, *DDX41*, *DNMT3A*, *EED*, *ELANE*, *ETNK1*, *ETV6*, *EZH2*, *FBXW7*, *FLT3*, *GATA1*, *GATA2*, *GFI1*, *GNAS*, *HNRNPK*, *HRAS*, *IDH1*, *IDH2*, *IKZF1*, *IL2RG*, *IL7R*, *KRAS*, *JAK2*, *JAK3*, *KDM6A*, *KIT*, *KMT2A*, *MAP2K1*, *MPL*, *NF1*, *NOTCH1*, *NPM1*, *NRAS*, *PAX5*, *PHF6*, *PIGA*, *PML*, *PRPF40B*, *PTEN*, *PTPN11*, *RAD21*, *RARA*, *RUNX1*, *SETBP1*, *SF1*, *SF3A1*, *SF3B1*, *SH2B3*, *SMC1A*, *SMC3*, *SRSF2*, *STAG1*, *STAG2*, *STAT3*, *STAT5A*, *STAT5B*, *SUZ12*, *TERC*, *TERT*, *TET2*, *TP53*, *U2AF1*, *U2AF2*, *WT1,* and *ZRSR2.* All coding exons for each gene were covered with an analytical sensitivity of 5% mutant reads in a background of wild-type reads. Established bioinformatics pipelines were used to identify somatic variants.

### 2.6. HTG Whole Transcriptome Sequencing and Profiling

Whole transcriptome sequencing was performed on archival FFPE tissue sections utilizing extraction-free HTG EdgeSeq technology (HTG Molecular Diagnostics, Inc., Tucson, AZ, USA). This platform shows high correlation with other RNAseq platforms and a low failure rate with FFPE [12,13]. The HTG whole transcriptome panel contains a total of 19,616 probes, including 4 positive control probes, 100 negative control probes, 22 genomic DNA probes, and 92 external RNA control consortium (ERCC) probes. The processing procedures combined quantitative nuclease protection assay technology with NGS on the Illumina NextSeq 500/550 sequencing platform. The HTG EdgeSeq Parser was used to align the FASTQ files to the probe list to collate the data. The quantitative data were provided as raw, QC (quality control) raw, log2 CPM (counts per million), and median normalized data. QC of RNA sequencing data was performed in line with the recommendations from HTG to meet cutoff requirements for sample quality, sufficient read depth, and minimal expression variability across probes. Samples that failed post-sequencing QC were excluded from analysis. Differential expression analysis of RNA expression data was performed using HTG EdgeSeq Reveal software (a web-based data analysis software), including the DESeq2 package (version 1.30.1) for differential expression analysis.

### 2.7. Statistical Analysis

Statistical analyses were performed using the Graph-Pad Prism 9 (San Diego, CA, USA) and SPSS 26.0 software (IBM Corporation, Armonk, NY, USA). Differences in the pathologic features between blastoid-HGBL and B-ALL cases were analyzed using the Fisher exact test. The continuous data were compared using one-way analysis of variance (ANOVA). A *p* value of less than 0.05 was considered statistically significant.

## 3. Results

### 3.1. Baseline Clinical Characteristics

Patients with blastoid HGBL included 77 men and 44 women with a male to female ratio of 1.75 and a median age of 59 years (range, 18–87 years) at initial diagnosis. Eighty-one (67%) presented with de novo disease and forty (33%) patients had a documented prior or concurrent diagnosis of follicular lymphoma (*n* = 27) or other type of non-Hodgkin lymphoma. Thirty-seven patients presented with bone marrow-based disease and eighty-four (69%) patients had lymphadenopathy or other extramedullary presentation including intraabdominal mass, soft tissue, stomach, bone lesions, chest wall, central nervous system, abdominal wall, lung, liver and breast, pleural fluid, thyroid, colon, mediastinum, tonsil, and submandible, etc. At the time of presentation, 80% of patients had extra-nodal sites of involvement, including bone marrow and central nervous system involvement. An elevated serum lactate dehydrogenase (LDH) level was present in 84 of 100 (84%) patients. Most patients (90%) had Ann Arbor stage III or IV disease and 79 of 114 (69%) patients had a high international prognostic index (IPI) score of ≥3. Fifty of ninety-three (54%) patients had circulating lymphoma cells identified by morphologic examination and/or flow cytometric immunophenotypic analysis, either at time of initial presentation or during the disease course.

Based on genetic data, 66 (55%) cases carried *MYC* and *BCL2* and/or *BCL6* rearrangements: 39 cases with *MYC* and *BCL2* rearrangements (double-hit lymphoma, DHL), 23 cases with *MYC*, *BCL2*, and *BCL6* rearrangements (triple-hit lymphoma, THL), and 4 cases with *MYC* and *BCL6* rearrangements. There were 55 (45%) cases of HGBL not otherwise specified (NOS); 22 (40%) had isolated *MYC*-rearrangement (*MYC*-R), 9 of 38 (24%) had *BCL2*-rearrangement (*BCL2*-R), and 2 of 37 (5%) harbored *BCL6*-rearrangement (*BCL6*-R).

The 47 B-ALL patients included 30 men and 17 women with a male: female ratio similar to blastoid-HGBL patients. The median age of this group was 44 years (range 18–87). All patients presented with bone marrow disease, although extramedullary involvement as shown by imaging studies was frequent in up to 50% of patients, including spleen, liver, lymph nodes (20%), and cerebrospinal fluid (CSF). No patients had a history of B-cell lymphoma.

### 3.2. Comparison of Immunophenotypic Features

All the immunophenotypic and molecular cytogenetic features were similar for the 37 bone marrow-presented cases and the 84 cases with extramedullary presentation (*p* > 0.05 for all, results not shown), therefore they were combined together as one blastoid-HGBL group to compare with the B-ALL group in this study.

Although blastoid-HGBL and B-ALL cases shared similar morphologic features (Figure 1), their immunophenotypes were different (Table 1, Figure 1 and Figure 2).

By immunohistochemistry, MYC (84% vs. 24%) and BCL6 (77% vs. 16%) expression were significantly more frequent in blastoid HGBL cases than B-ALL (*p* < 0.0001, Table 1 and Figure 1). TdT was positive more frequently in B-ALL than in blastoid HGBL (89% vs. 17%) (*p* = 0.0001). TdT often showed strong expression in all or a large subset of lymphoblasts in B-ALL, in contrast with variable expression in a small subset of lymphoma cells in blastoid HGBL (Figure 1). The frequency of BCL2 overexpression was similar between blastoid-HGBL and B-ALL cases (*p* = 0.592), but staining intensity was more variable in blastoid HGBL compared with B-ALL cases, in which BCL2 was always strong and diffuse. Ki67 proliferation rate was greater than 80% in 75% of blastoid-HGBL cases. All cases tested were negative for cyclin D1.

Detailed flow cytometry immunophenotypic analysis was performed in 96 cases of blastoid HGBL and all cases of B-ALL. All cases were positive for one or more pan-B-cell antigens. Blastoid-HGBL cases had significantly more frequent CD45 expression at an intensity greater than that of granulocytes, dimmer CD10 expression relative to the level of hematogones, brighter CD20 comparable to mature B cells, and brighter CD38 at the level of hematogones (*p* < 0.01 for all) (Table 1 and Figure 2). Three-fourths of blastoid-HGBL cases were positive for monotypic surface light chain and all blastoid-HGBL tested were negative for CD13, CD15, CD33, CD34, and CD117. TdT was positive variably in 5 of 30 (17%) blastoid-HGBL cases. By contrast, B-ALL cases more frequently expressed TdT in 40 of 45 (89%)) and one or more myeloid antigens in 19 of 46 (41%) cases assessed (*p* = 0.0001). CD34 was positive in 22 of 47 (42%) B-ALL cases, but it should be noted that we purposely included a large subset of CD34 negative B-ALL to better serve the aim of this study. All five B-ALL cases assessed were negative for surface light chain.

Combining immunohistochemistry and flow cytometry results, TdT was assessed in a total of 87 cases of blastoid HGBL and 45 cases of B-ALL. TdT expression was significantly more frequent in B-ALL (89%, 40/45) than in blastoid HGBL (17%, 15/87). Twenty-three blastoid HGBL and seventeen B-ALL cases were examined by both flow cytometry and immunohistochemistry; twenty-one HGBL and all seventeen cases of B-ALL showed concordant results. The two discordant blastoid-HGBL cases had weak/dim positivity for TdT in a small subset of lymphoma cells by immunohistochemical analysis but were negative by flow cytometry. Therefore, flow cytometry and immunohistochemistry had a high concordance rate (95%).

### 3.3. Comparison of Cytogenetic Features

FISH analysis for *MYC* was detected in 87 (72%) cases of blastoid HGBL (Table 2), including 22 cases of blastoid HGBL-NOS with *MYC*-R alone and 65 cases of blastoid D/THL carrying concurrent *BCL2* and/or *BCL6* rearrangements. Of the 15 TdT+ cases of blastoid HGBL, 10 were D/THL and 5 were HGBL-NOS, the latter group including 3 with *MYC*-R and 2 with extra copies of *MYC*.

Conventional karyotyping showed a complex karyotype in 53 of 57 (93%) cases of blastoid HGBL and 19 of 42 (45%) B-ALL cases (*p* = 0.0001; Table 2). The four cases of blastoid HGBL with a non-complex karyotype included one DHL and three HGBL-NOS.

In B-ALL cases, none of the 44 tested cases had *MYC*-R as shown by karyotyping and/or FISH. Seven cases carried *BCR::ABL1,* of which all were CD34+; ten cases had *MLL*-R and another ten cases harbored t(1;19)/*TCF3::PBX1* or translocations involving *TCF3* (*E2A*) on chromosome 19p13 with other partner genes; these twenty cases were CD34 negative. B-ALL related translocations were not identified in the blastoid-HGBL cases.

### 3.4. Comparison of Gene Mutations by Targeted NGS

Targeted NGS was performed in 25 cases of blastoid HGBL and 41 cases of B-ALL (Table 2). The most frequently mutated gene in blastoid-HGBL cases was *TP53* in 11 (44%), followed by *CREBBP* in 5 (20%). Notably, all 11 *TP53*-mutated blastoid HGBL cases carried *MYC*-R, including 4 HGBL-NOS, 5 DHL, and 2 THL. *TP53* mutation was less common (6/41; 15%) in B-ALL cases (*p* = 0.018). Mutations in *NRAS* and *KRAS* were seen exclusively in B-ALL patients (22% and 17%, respectively), whereas none of these mutations were detected in blastoid-HGBL cases (both *p* < 0.05).

### 3.5. Validation of the Previously Reported Six-Point Flow Cytometry-Focused Scoring System

Our previous study established a scoring system based on the assessment of six parameters (one point each) focused on flow cytometric staining patterns for CD45, CD10, CD20, CD38, and TdT expression and *MYC*-R or MYC overexpression [4]. A score of three or above supported a diagnosis of blastoid HGBL, whereas a score of <3 favored B-ALL. We assessed this scoring system in 92 blastoid-HGBL cases with all six markers tested, including 33 cases in bone marrow and 59 in lymph node or extranodal sites other than bone marrow. All 92 cases had a score ≥3, including 69 cases with a score of ≥4 and 23 cases with a score of three (Table 3). Forty-six cases of B-ALL had complete flow cytometry evaluation and forty cases had a score of one or two supporting the diagnosis; the remaining six cases had a score of three (Table 3). In these six cases, one was CD34+, therefore was definitive for B-ALL. The other five cases were CD34-negative and all had the myeloid marker expression, including three cases with recurrent translocations involving *MLL*, supporting a diagnosis of B-ALL. Overall, the sensitivity, specificity, positive predictive value, and negative predictive value of this six-point scoring system in the 59 non-bone marrow blastoid-HGBL-NOS cases were 100%, 87%, 91%, and 100%, respectively, and in the 33 bone marrow cases were 100%, 87%, 85%, and 100%, respectively.

### 3.6. Comparison of the Two Scoring Systems

Ninety-two cases of blastoid HGBL had complete data for the six-point flow cytometry-based scoring system, of which sixty-four had complete data for the three-point IHC-based scoring system, allowing for comparison in sixty-four blastoid-HGBL cases. As shown in Table 3, all 64 cases of blastoid HGBL had a score of ≥3 in the six-point scoring system, but 5 blastoid-HGBL cases had a misleading score of <2 in the three-point IHC-based scoring system. Similarly, the two scoring systems can be compared in 37 cases of B-ALL, 8 of which had a misleading score, including 5 with a misleading score in the six-point scoring system, 2 cases in the three-point IHC-based scoring system, and 1 in both systems with a score that suggested a diagnosis of blastoid HGBL. Overall, 13 of the 101 cases (64 blastoid HGBL and 37 B-ALL) had a misleading score and in 12 cases there was a discrepancy between the two scoring systems. The B-ALL case with concordant misleading scores in both systems expressed CD34 and was definitive for B-ALL diagnosis.

### 3.7. Validation by Whole Transcriptome Profiling and Establish a Diagnostic Algorithm

RNASeq-based whole transcriptome profiling was performed, utilizing the HTG EdgeSeq technology on a subset of cases. Differential expression results have been reported previously [5]. Comparing cases with concordant scores by the two scoring systems, blastoid-HGBL cases showed a distinctive gene expression profile compared with B-ALL, as demonstrated by the top 40 differentially expressed genes (Figure 3). Nine of thirteen cases with misleading scores by the two scoring systems had HTG gene expression study performed. The gene expression profile confirmed that four cases (three CD34-negative) of B-ALL with a six-point score of three and three cases of B-ALL with a three-point score of two (one of which also had a score of three in the six-point system) all had gene expression signatures similar to the B-ALL group, confirming the diagnosis of B-ALL (Figure 3). The diagnosis of the three cases of blastoid HGBL with a three-point score of <2 was also confirmed by the gene expression signature (Figure 3).

We further analyzed gene expression levels for markers significantly differentially expressed at the protein level between blastoid HGBL and B-ALL in the two scoring systems (Figure 4A). *DNTT* (the gene encoding TdT) was not included in the HTG whole transcriptome panel. As shown in Figure 4A, genes encoding *MYC*, *BCL6*, *CD20,* and *CD45* were all upregulated significantly at the mRNA level at different fold changes compared with B-ALL (*p* < 0.05 for all). *MME* (*CD10*) was downregulated in blastoid HGBL in comparison with B-ALL (*p* < 0.05). mRNA for CD38 showed 1.4-fold upregulation in blastoid HGBL (*p* = 0.13). Of note, *MYC* was upregulated by 1.85-fold in blastoid HGBL (*p* = 0.0024, Figure 4A) and was associated with the activation of *MYC*-targeted molecules such as *CEBPB* and *E2F1*. The upregulation of *MKI67* and *PCNA* by 1.7 and 1.87-fold, respectively, in blastoid HGBL is in keeping with the morphologic findings of brisk mitotic activity and high proliferation in blastoid HGBL, providing a molecular basis for the aggressive features observed clinically and pathologically.

Although as a whole group, genes encoding MYC, BCL6, CD20, and CD45 were all upregulated significantly and *MME* (CD10) was downregulated compared with B-ALL; the gene expression level of these five genes in the nine cases with misleading scores showed unusual results (Figure 4B). Compared with those cases with unequivocal supporting scores, these cases with misleading scores showed variable lower expression of *MS4A1* (CD20) and *BCL6* and increased *MME* (CD10) in the three cases of blastoid HGBL, mimicking B-ALL. In contrast, the six cases of B-ALL with misleading scores showed greater variations but relatively increased expression in *MYC*, *MS4A1* (CD20), *CD38*, and *PTPRC*(CD45) and lower expression of *MME* (CD10) compared with typical B-ALL cases. All these changes contributed to their misleading scores at both protein and mRNA levels. Despite dissecting out the molecular basis underlying the discordant scores, the overall gene expression patterns in these challenging cases were confirmatory for their diagnosis in each category (Figure 3).

The accuracy of the six-point flow cytometry-based scoring system and the three-point IHC-based scoring system were 96% and 92%, respectively. If these two scoring systems are combined, the accuracy of a concordant score for classification of a blastoid B-cell neoplasm is 99% (88/89). For diagnostically challenging blastoid B-cell neoplasm, if the two scores are concordant, a diagnosis can be made. If the two scores are discordant, then these are very challenging cases and further correlation with all available clinical and genetic features is needed, as shown in the diagnostic algorithm incorporating the two scoring systems (Figure 5). For example, a 73-year-old female was found anemic with 38% atypical lymphoid cells in peripheral blood; a bone marrow biopsy showed diffuse infiltration of medium-sized blastoid cells. Flow-cytometric analysis showed these cells were positive for CD45 (decreased, <granulocytes), CD10 bright, CD19, CD20, CD22, and CD38 but negative for CD34, TdT, and surface light chains. Immunostains showed the cells were positive for BCL6 but negative for MYC and TdT. A lack of surface light chain and decreased CD45 favor B-ALL; however, both CD34 and TdT were negative and up to 30% of large B-cell lymphoma lack surface light chain expression. Expression of CD20 and BCL6 favors HGBL, however bright CD10 and decreased CD45 do not. Therefore, both the morphology and immunophenotype were not definitive for either blastoid HGBL or B-ALL. Weeks later, molecular cytogenetic studies showed the loss of one copy of *TP53* and no lymphoma or B-ALL-defining translocations or other aberrancy. Therefore, this case was still difficult to classify even with all combined results. Using our scoring systems, the case had a score of two in the six-point system and a score of one in the three-point system, concordantly supporting the diagnosis of B-ALL. A further transcriptome profile for research purposes also confirmed the diagnosis of B-ALL. The patient received B-ALL-related treatment and was MRD negative for more than one year up to last follow-up.

## 4. Discussion

This study aimed to improve our understanding of the differential diagnosis of blastoid B-cell neoplasm. Since blastoid mantle cell lymphoma is relatively easier to diagnose based on expression of cyclin D1 or/and *CCND1* rearrangement, we focused on the differential diagnosis of blastoid HGBL, either DHL or NOS, versus B-ALL. Distinguishing blastoid HGBL from B-ALL is important since the treatment and prognosis of patients with these neoplasms differ substantially. To achieve our aims, we validated the previously proposed six-point flow cytometry-focused scoring system in a large cohort with mostly non-bone marrow cases. We also compared the six-point scoring system with a three-point IHC-based system and identified very challenging cases with discordant scores by the two scoring systems. The diagnosis of these challenging cases was further assessed by gene expression profiling.

In general, lymphoma is a nodal-based disease that often occurs in adults, whereas B-ALL is a bone marrow-based disease and frequently affects children. However, blastoid HGBL often presents and/or involves bone marrow, whereas up to 25% of B-ALL cases occur in adults, with extramedullary presentations in a subset of cases that often cause diagnostic challenges [14,15,16]. In a study by Moore et al., blastoid-HGBL cases were found to be enriched in bone marrow specimens as well as in the D/THL category [17]. We previously reported 31 cases of blastoid HGBL with an initial presentation in bone marrow [4]. In this study, bone marrow presentation with or without circulating lymphoma cells was common, present in ~27% of patients with blastoid HGBL in this larger cohort, mimicking the clinical manifestations of B-ALL patients. Although most B-ALL cases are CD34+ and easy to recognize, up to a third of B-ALL may lack CD34 expression, making the differential diagnosis with blastoid HGBL difficult [18,19].

The diagnostic challenge not only lies in the similar blastoid morphology of the neoplastic cells, but the overlapping immunophenotypic features between blastoid HGBL and B-ALL. A subset of blastoid-HGBL cases shows an aberrant immunophenotype that can mimic B-ALL, such as dim to negative CD20, decreased CD45, TdT expression, and absence of surface light chains, as shown in this study. On the other hand, B-ALL cases may show unusual features such as lack of CD34, bright CD38, and decreased or absent CD10, particularly in B-ALL patients harboring translocations involving *MLL* and *TCF3*. A mature B-cell immunophenotype also has been occasionally identified in rare cases of B-ALL [20,21] and in a few cases in this study. Therefore, no single marker can distinguish blastoid HGBL from B-ALL with the exception of CD34. CD34 expression, even in a small subset of blasts, is diagnostic of B-ALL and no further differential diagnosis is needed. A scoring system simultaneously evaluating multiple markers is helpful for guiding the work-up and establishing a correct diagnosis. For the purpose of the correct diagnosis of these difficult cases, we have previously reported two scoring systems, one flow cytometry-focused six-point scoring system and one immunohistochemistry-focused three-point scoring system [4,5].

The six-point flow cytometry-focused scoring system was developed in blastoid-HGBL cases presenting in bone marrow; it is unclear whether it also has a high performance in extramedullary cases. In this study, we further evaluated the immunophenotype and molecular cytogenetic features in a larger cohort of 121 cases of blastoid HGBL diagnosed in bone marrow and more often extramedullary sites. We showed that blastoid-HGBL cases have distinctive immunophenotypic, molecular, and cytogenetic features, including more frequent expression of brighter CD45, CD20, CD38, BCL6, and MYC overexpression and less frequent bright CD10 and TdT expression. Blastoid-HGBL cases also more frequently have *MYC* rearrangement, a complex karyotype, and *TP53* aberrancies, whereas myeloid marker expression and *KRAS* and *NRAS* mutations occur exclusively in a subset of B-ALL cases. Based on these distinctive features, we further evaluated the performance of the six-point score system in extramedullary blastoid-HGBL-NOS cases and compared them with those bone marrow cases. The results confirmed that the six-point flow cytometry-focused scoring system has a similar performance in both bone marrow and extramedullary blastoid-HGBL subgroups, supporting that the six-point scoring system had a good performance regardless of tissue origin.

In previous studies of blastoid HGBL in comparison with B-ALL, we established a six-point flow cytometry-focused scoring system and a three-point immunohistochemistry-focused scoring system for the differential diagnosis of blastoid HGBL from B-ALL with excellent performance [4,5]. Both scoring systems were further validated by RNA-seq-based gene expression profiling in a subset of cases with available material, confirming their correct classification by using the two scoring systems. The molecular basis for the distinctive immunophenotypic features used in the two scoring systems was confirmed at a transcriptional level in this study. Using the six-point scoring system, all 92 cases of blastoid HGBL had a score of at least three with a sensitivity of 100%. Six of forty-six B-ALL cases also had a score of three, which would be falsely classified as lymphoma. Using the three-point IHC-focused scoring system, eight cases would be falsely classified, including five cases of blastoid HGBL with a score of one and three B-ALL cases having a score of two. One of the B-ALL cases had misleading scores in both scoring systems. Therefore, a total of 13 cases (5 blastoid HGBL and 8 B-ALL, Table 3) had misleading scores. The accuracy of the previously defined six-point scoring system was 96%, whereas the accuracy of the three-point score system was 92%. For the cases of blastoid HGBL (*n* = 64) and B-ALL (*n* = 37) with data sufficient to assess both scoring systems, the concordance rate was 88%. Twelve of thirteen cases with misleading scores had discrepant scores by the two different scoring systems. This result suggests that using both scoring systems together improves the accuracy of classification of blastoid B-cell neoplasms to 99%. Comparison of the two scoring systems and the scores of the 13 cases with misleading scores suggests that a score of ≥4 or ≤2 in the six-point score system and a score of three or zero in the three-point score system are more definitive for classification. For cases with a score of three in the six-point system or a score of one or two in the three-point system, concurrent use of both scoring systems was helpful. If the scores were concordant, the classification was firm. Cases that had discordant scores between the two scoring systems were extremely challenging and required correlation with all available clinicopathologic features. When correlated with a history of non-Hodgkin lymphoma, expression of myeloid markers, complex karyotype, *TP53* gene aberrancies, *KRAS* and *NRAS* mutations, and B-ALL associated translocations (Figure 5), all 13 cases were correctly classified. The classification of these difficult cases was further confirmed by gene expression profiling.

Six (16%) cases of B-ALL had BCL6 overexpression and two (33%) had t(1;19) or *TCF3* rearrangement. Similar findings were reported in a case series of 52 B-ALL cases, in which 12% showed BCL6 expression; half of these cases were strongly BCL6 positive and carried t(1;19)/*TCF3(E2A)::PBX1* [22]. Conversely, TdT expression was detected in about 20% of blastoid-HGBL cases and was seen exclusively in blastoid-HGBL cases with *MYC* abnormalities in this study. This observation was similar to findings in a recent report in which TdT expression was associated with a constellation of clinicopathologic features, including *MYC*-R, blastoid morphology, GCB-phenotype, uniform CD34-negativity, and a poorer prognosis [2].

## 5. Conclusions

In summary, blastoid-HGBL cases have distinctive immunophenotypic, molecular, and cytogenetic characteristics compared with B-ALL. Concurrent use of both a six-point flow cytometry-focused and a three-point IHC-focused scoring system improves the accuracy of classification of blastoid B-cell neoplasms. Cases that had discordant scores between these two scoring systems were extremely challenging and correct classification of them required correlation with all available clinicopathologic features.

## Figures and Tables

**Figure 1 cancers-15-00848-f001:**
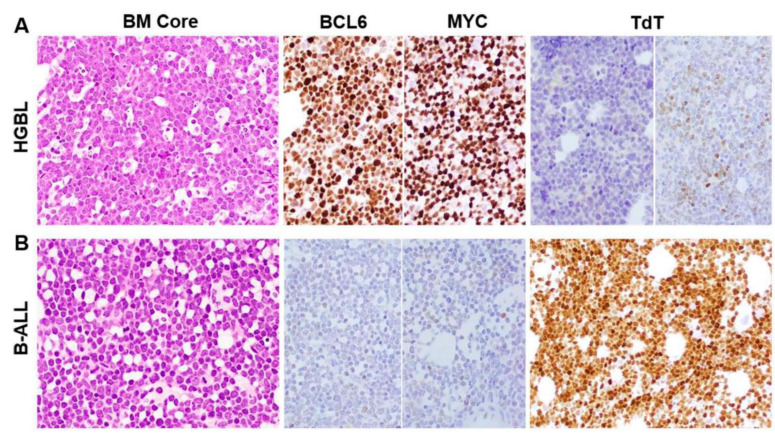
Comparison of morphologic and immunohistochemical features between blastoid HGBL and B-ALL. A representative case of blastoid HGBL (**A**) showed marrow space being entirely replaced by blastoid lymphoma cells with a starry sky pattern and strongly and diffusely positive for BCL6 and MYC but negative for TdT (left). Occasional blastoid HGBL may be variably positive for TdT in a small subset of lymphoma cells (right). In contrast, the lymphoblasts of a representative B-ALL case (**B**) showed similar blastoid morphology on core biopsy but had an opposite immunophenotypic profile featured by negative BCL6 and MYC and strong diffuse positive TdT. Magnifications: H&E, ×50; immunostains, ×40.

**Figure 2 cancers-15-00848-f002:**
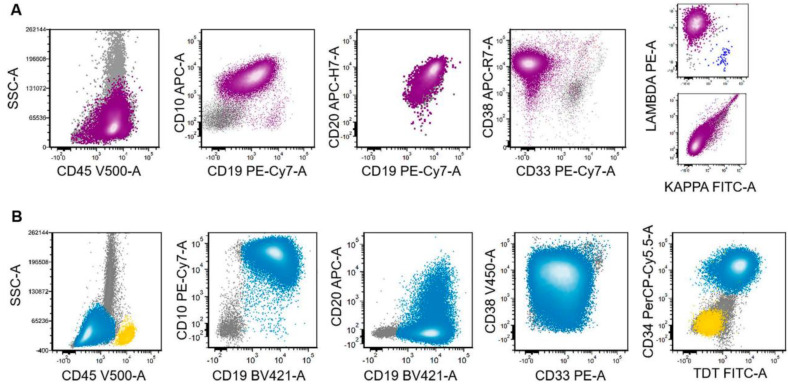
Comparison of immunophenotypic features in blastoid HGBL versus B-ALL by flow cytometric analysis. (**A**) A representative case of blastoid HGBL (magenta) showed brighter CD45, less bright CD10, relatively stronger CD20, and brighter CD38 with or without (small subset cases) surface light chains. (**B**) In contrast, the lymphoblasts (blue) of B-ALL showed the opposite expression patterns for these markers, in addition to the expression of myeloid antigens (CD33) and immature markers (CD34 and TdT). Yellow population indicates the normal lymphocytes and gray for granulocytes in the background.

**Figure 3 cancers-15-00848-f003:**
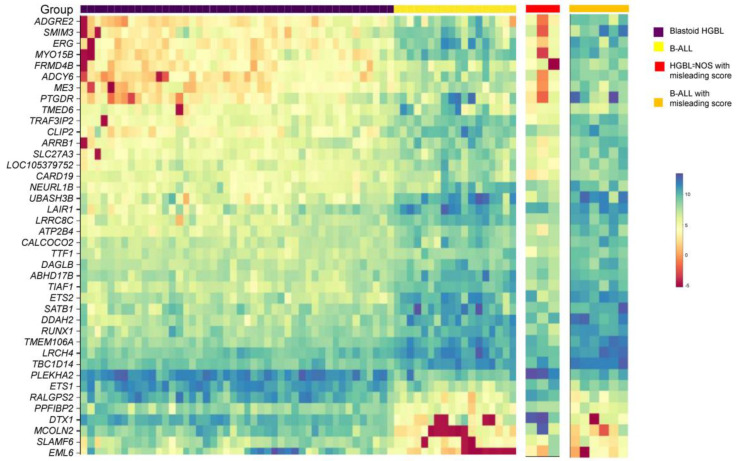
Transcriptome profiling confirmed the correct diagnosis of cases with misleading scores. Heatmap of the top 40 differentially expressed genes for blastoid HGBL in comparison with B-ALL. The 3 cases of blastoid HGBL and 6 cases of B-ALL with misleading scores showed a gene expression profile similar to blastoid HGBL and B-ALL, respectively.

**Figure 4 cancers-15-00848-f004:**
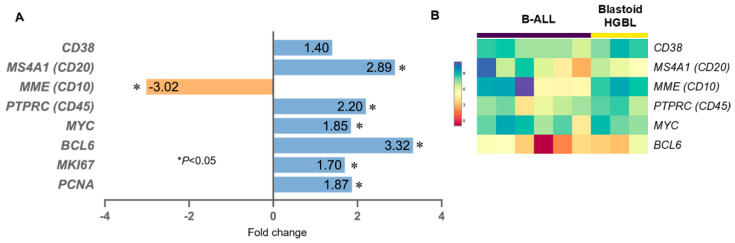
Comparison of the mRNA expression level of the six genes included in the two scoring systems between blastoid HGBL versus B-ALL. (**A**) Average gene expression level of blastoid HGBL in comparison with B-ALL. The two genes encoding Ki67 and PCNA for proliferation rate assessment are also included for comparison. (**B**) Heatmap of these 6 genes for the 9 cases with misleading scores in the two scoring systems.

**Figure 5 cancers-15-00848-f005:**
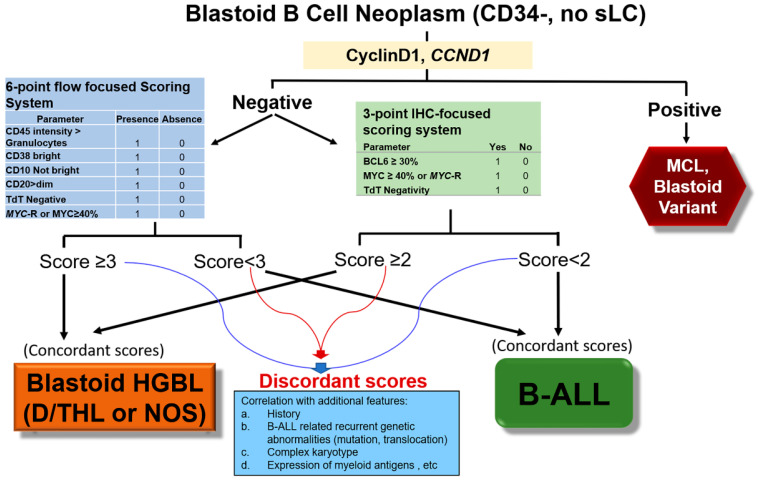
Recommended algorithm for the differential diagnosis of blastoid B-cell neoplasm, HGBL, high-grade B-cell lymphoma, B-ALL, B-acute lymphoblastic leukemia/lymphoma, MCL, mantle cell lymphoma. The thin blue and red lines were used for different combinations for discordantly scored cases between the two scoring systems.

**Table 1 cancers-15-00848-t001:** Comparison of immunophenotyping between blastoid HGBL and B-ALL.

Markers	Blastoid HGBL % (Positive/Total)	B-ALL % (Positive/Total)	*p* Value
Flow cytometry			
CD45 intensity > granulocytes	75 (72/96)	23 (11/47)	**<0.0001**
CD10 not bright (<hematogones)	86 (83/96)	58 (27/47)	**0.0001**
CD19	100 (96/96)	100 (47/47)	1.000
CD20 intensity ≥ mature B cells *	72 (47/65)	11 (5/46)	**0.0001**
CD22	92 (88/96)	98 (44/45)	0.104
CD34	0 (0/96)	47 (22/47)	**<0.0001**
CD38 bright (≈hematogones)	70 (67/96)	38 (18/47)	**0.0005**
Surface light chains	75 (72/96)	0 (0/5)	**0.002**
TdT negative **	83 (72/87)	11 (5/45)	**<0.0001**
Myeloid markers (CD33, etc.)	0 (0/30)	41 (19/46)	**0.0001**
Immunohistochemistry			
MYC IHC ≥ 40%	84 (78/93)	24 (9/37)	**<0.0001**
BCL2 IHC ≥ 50%	85 (98/115)	100 (7/7)	0.592
BCL6 IHC ≥ 30%	77 (84/109)	16 (6/37)	**<0.0001**

* CD20 positivity was assessed prior to rituximab-based chemotherapy; ** TdT was evaluated using either flow cytometry and/or immunohistochemistry. Bold: statistically significant.

**Table 2 cancers-15-00848-t002:** Cytogenetic and molecular comparison between blastoid HGBL versus B-ALL.

Markers	Blastoid HGBL *n* = 121 % (Positive/Total)	B-ALL *n* = 47 % (Positive/Total)	*p* Value
Cytogenetics			
*MYC*-R	72 (87/121)	0 (0/44)	**<0.0001**
Complex karyotype	93 (53/57)	46 (19/41)	**0.0001**
B-ALL associated translocations	5 (0/57)	57 (27/47)	**0.0001**
Somatic mutations by NGS			
*TP53*	44 (11/25)	15 (6/41)	**0.018**
*CREBBP*	20 (5/25)	7 (3/41)	0.242
*NRAS*	0 (0/25)	22 (9/41)	**0.011**
*KRAS*	0 (0/25)	17 (7/41)	**0.039**
*JAK1/2/3*	0 (0/25)	7 (3/41)	0.283

Bold: statistically significant.

**Table 3 cancers-15-00848-t003:** Comparison of the two scoring systems in Blastoid HGBL and B-ALL.

	6-Point Score		3-Point Score		No. of Discordant Cases	
	≥3	<3	≥2	<2	Total	with HTG
Blastoid HGBL (*n* = 64)	64	0	59	**5**	**5**	3
B-ALL	**5**			5		
(*n* = 37)	**1**		**1**			
		2	**2**			
		29		29	**7**	6

Bold: cases with misleading scores.

## Data Availability

Data are available for sharing upon reasonable request to the corresponding author.

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
