# Peer review of "Blastoid B-Cell Neoplasms: Diagnostic Challenges and Solutions"

_cancers, 2023, doi:10.3390/cancers15030848_

Round 1

Reviewer 1 Report

This study is an extension of the 2 prior published studies from this group that established a 6-point flow score and 3-point IHC score, respectively, for the discrimination of HGBL (n=121) from BALL (n=47).  The novelty of this study is in the application of these scores to non-bone marrow tissues, a comparison of the scoring systems, and the use of transcriptomic methods to examine the cases where the two scoring systems were discordant.

Major issues: 

1. This study does not adequate explain how this scoring system would be used in clinical practice.  In fact, the logic of the design seems a bit circular. The authors have selected cases where the definitive diagnosis was presumably made using flow, IHC, and other molecular/genetic data and that this diagnosis is presumably the “gold standard.”  Then the authors apply their score and state that flow and IHC scores can distinguish the two diagnoses except in cases where they also need molecular/genetic data... A better description of what was used as the gold standard for classification in these cases is needed.  Then some result needs to show how using the scoring system improves on the diagnosis.  Instead, the conclusion appears to be that you need flow and/or IHC and molecular/genetic data to make the diagnosis, which is how the diagnoses were initially made. 

a. Page 10 lines 341-347: If these 9 cases had misleading scores AND misleading transcriptomic profiles, then the gold standard is the original morphologic and immunophenotypic (and cytogenetic) studies.  So, what is the added value of having these scores if they do not add to the diagnostic accuracy of the standard of care process?

b. Page 11 line 425-427: Please explain how the fact that there were 12/13 cases with discordant scores would indicate that combining the scoring systems would perform better. These discordances appear to identify cases where even the transcriptomic studies do not help distinguish the two diagnostic possibilities, so how does this help?

c. Page 10 Figure 5 and page 12 lines 450-452 (concluding sentence): Are the authors suggesting that they would not wait for cytogenetic (FISH or karyotype) and possibly molecular studies in making a diagnosis of HGBL and B-ALL? Please clarify the utility in using these scores in the diagnostic setting. Would the authors therefore be able to bin the cases as HGBL or B-ALL without waiting for the cytogenetic studies? What do the thin red and blue lines going to the discordant score box indicate?

2. Did the authors do the same analysis on the extramedullary samples and compare to the bone marrow samples? The data appears to have only examined them in aggregate. That comparison would be indicated to address their aim #1.  Without this comparison, page 11, line 405-407 cannot be stated.

3. Page 2, line 86: Previously reported cases were included from reference 4 and 5 only for the CD34 negative BALL cases (25/47)?  Were they part of the cohorts used to validate the 6-point flow and 3-point IHC scores? These samples, if used in the validation of the score, should be excluded.

Minor issues:

1. Some grammatical and syntactical errors are found in the text.

a. Page 1 line 18-19: “Comparing the two scoring systems...” should be “A comparison of the two scoring systems... showed a concordance”

b. Page 2 line 62 and 64 and 91:  two spaces between system and in, between potential and solutions, between aspirates and as  (probably other examples in the text)

c. Page 4 line 168: missing the total N for the HGBL cases

d. Page 4 line 148: missing a period

e. Page 9 line 322: should read “DNTT, the gene encoding terminal deoxynucleotidyl transferase (TdT), was not included...” (“the” gene, not “a” gene). 

f. Page 10 line 345: the representation of genes and proteins is inconsistent [i.e., MS4A1 (CD20) but PTPRC/CD45]

g. Page 9 line 329: Ki67 and PCNA are not shown in Figure 4 (or in Figure 3). This data should be shown in Figure 4 or at least in the supplemental since reference is made to their clinical implications.  In addition, Ki67 is incorrectly denoted. The gene (which would then be italicized) should be MKI67 while the protein is commonly referred to as Ki67 and should not be italicized.

h. Page 11 line 374: et al. requires italicization and the period since it is an abbreviation of the Latin et alia.

i. Page 11, line 392: spelling error- “sunset” should be “subset”

2. Define abbreviations:

a. Page 4 line 185-186: define MYC-R, BCL2-R and BCL6-R as rearrangements of those genes

3. Page 4, section 3.1 “Baseline clinical characteristics”: since the point of the paper is to validate the scoring systems in different tissues other than bone marrow, it would be good to have a table of the clinical presentations including a list of the various extramedullary sites of involvement and which sample sites were included in the analysis.

4. Page 8 line 291: Please clarify that the 92 cases are of HGBL (as lines 293 as well as 295-296 would imply). 

5. Page 8 line 298-299: this sentence is somewhat confusing. It would be clearer to state that “13 of 101 cases (64 HGBL and 37 B-ALL) had a misleading score...”

6. Page 10, Figure 5: Do the authors not include SOX11 in their work-up for blastoid mantle cell lymphomas (see PMID: 22251940)?  Also, this is mentioned on page 10 line 361 and page 1 lines 36-37.

7. Are there any clinical outcomes that would clarify the diagnoses of the borderline cases?

Reviewer 2 Report

Could athors explain a little bit further what the golden standard of case selection was, which is the essence on which the scoring system is based?

how do we know for sure cases have not been mixed at the inclusion process?

Where do the selected cases cluster with RNAseq? According to their assigned group?

Intro could be written more consised.

Round 2

Reviewer 1 Report

This revision of the original submission addresses some of the issues with the first version. However, the authors have in many cases NOT addressed the scientific design issues of their studies or the persistently contradictory language.

1)       The authors have clarified in the methods that they lumped the BM and the extramedullary cases.  However, again, that precludes the authors making any statement of that the scoring system has a similar performance regardless of tissue of origin. This a basic concept in scientific method. You have to actually compare the performance in unadulterated arms.  Therefore, the extramedullary has to be analyzed separately from the BM cases if the authors want to make these claims.  

2)       The transcriptomic data is presented in a confusing manner (at least to this reviewer...). In the rebuttal letter, the authors state that they did NOT have any misleading transcriptomic profiles and the first paragraph (and figure) delineates the 9 cases.  Then the third paragraph in that section points out the changes that “contributed to their misleading scores to both protein and mRNA level.”  This appears to be contradictory and was the cause for this reviewer’s confusion – the text LITERALLY says that the mRNA score is misleading. Please clarify that, although the transcriptomic signature was more subtle in these cases, it was still definitive.

3)       Figure 5 should have a legend that defines the red and blue squiggly arrows.  It is not sufficient to define them in the rebuttal letter. They need to be defined in the figure. (And, why is one pair red and other blue? You also have blue straight lines.  Those should be black.) Please note that there is no Figure 5B, either, although it is referenced in the text

The manuscript would still benefit overall from a clear reframing of the purpose of the study.  The cosmetic changes and highlighted text do not clearly make the points that as the authors intend.

4)       The authors need to clearly indicate that the utility of the scoring system is to obtain the correct diagnosis within the first 1-2 days from just immunophenotypic results rather than waiting for cytogenetic or molecular results, which in some institutions make take longer.  The only comment is the “right after biopsy” added to the penultimate sentence.  That does not make the value of the approach clear until the end of the paper.

5)       The entire premise of the study is that distinguishing blastoid HGBL from BALL is challenging.  Therefore, although the authors state in the manuscript that they apply WHO criteria as the gold standard, in real life, there are always cases when the WHO criteria still leaves the pathologist in a diagnostic quandary (as the authors mention in their rebuttal letter).  Therefore, the WHO is NOT a gold standard in these challenging cases and those are the cases that the topic sentences of the manuscript seem to be implying that the authors will address.  Therefore to have the rebuttal letter say that they of course just used the WHO significantly diminishes the excitement for this scoring system.  The authors need to state very clearly that the study does not directly address these tough diagnoses, but instead provides rapid support for those cases that fall clearly within the WHO classification scheme before the molecular/cytogenetic results are available. It would be nice to have some really tough cases included in the mix and see what the scoring system comes up with or, in the absence of doing more work, at least acknowledging that this paper does not actually address those cases, but that one might hypothesize that the scoring system MIGHT help with those cases.

Not all of the minor grammatical issues are corrected, for instance d. Page 4 line 148: missing a period – is still missing the period.  Please recheck for typos.

Reviewer 2 Report

No new comments.

Round 3

Reviewer 1 Report

Thank you for the corrections and clarifications. No further comments.